# Continuous U-Net: Faster, Greater and Noiseless

**Chun-Wun Cheng** *                                             *cwc56@cam.ac.uk*
*Department of Applied Mathematics and Theoretical Physics, University of Cambridge*

**Christina Runkel** *                                           *cr661@cam.ac.uk*
*Department of Applied Mathematics and Theoretical Physics, University of Cambridge*

**Lihao Liu** *                                                  *ll610@cam.ac.uk*
*Department of Applied Mathematics and Theoretical Physics, University of Cambridge*

**Raymond H Chan**                                               *raymond.chan@cityu.edu.hk*
*Department of Mathematics, City University of Hong Kong*
*Hong Kong Centre for Cerebro-Cardiovascular Health Engineering*

**Carola-Bibiane Schönlieb**                                     *cbs31@cam.ac.uk*
*Department of Applied Mathematics and Theoretical Physics, University of Cambridge*

**Angelica I Aviles-Rivero**                                     *ai323@cam.ac.uk*
*Department of Applied Mathematics and Theoretical Physics, University of Cambridge*

**Reviewed on OpenReview:** *https://openreview.net/forum?id=ongi2oe3Fr*

## Abstract

Image segmentation is a fundamental task in image analysis and clinical practice. The current state-of-the-art techniques are based on U-shape type encoder-decoder networks with skip connections called U-Net. Despite the powerful performance reported by existing U-Net type networks, they suffer from several major limitations. These issues include the hard coding of the receptive field size, compromising the performance and computational cost, as well as the fact that they do not account for inherent noise in the data. They have problems associated with discrete layers, and do not offer any theoretical underpinning. In this work we introduce continuous U-Net, a novel family of networks for image segmentation. Firstly, continuous U-Net is a continuous deep neural network that introduces new dynamic blocks modelled by second order ordinary differential equations. Secondly, we provide theoretical guarantees for our network demonstrating faster convergence, higher robustness and less sensitivity to noise. Thirdly, we derive qualitative measures to tailor-made segmentation tasks. We demonstrate, through extensive numerical and visual results, that our model outperforms existing U-Net blocks for several medical image segmentation benchmarking datasets.

## 1 Introduction

Image segmentation is a fundamental task in medical image analysis and clinical practice. It is a critical component in several applications including diagnosis, surgery-guided planning and therapy. Manual segmentation of such medical datasets is time-consuming, and increases financial cost. The advent of deep learning, and in particular Fully Convolutional Neural Networks (FCNNs) (Long et al., 2015), opened the door to automatic segmentation techniques. The current state-of-the-art techniques are based on U-shape type encoder-decoder networks with skip connections (Qin et al., 2020; Liu et al., 2020; Zhou et al., 2019;

---

*Equal Contribution.

Su et al., 2021; Valanarasu & Patel, 2022). This family of networks has demonstrated astonishing results due to their representation learning capabilities, and the ability to recover fine-grained details.

The seminal paper of Ronneberger et al. (Ronneberger et al., 2015) introduced the U-Net model for Biomedical Image Segmentation. More precisely, U-net seeks to capture both the context and the localisation features. Firstly, it uses skip connections to provide additional information that helps the decoder to generate better semantic features. Secondly, it consists of a symmetric encoder-decoder scheme, which reduces the computational cost. The impressive performance of U-Net motivated the fast development of several U-Net variants e.g., (Qin et al., 2020; Su et al., 2021; Valanarasu & Patel, 2022; Etmann et al., 2020), and its usage in wide variety of clinical data.

Despite the powerful performance reported by existing U-Net type networks, *they suffer from several major limitations.* ● Firstly, they hard code the receptive field size, which requires architecture optimisation for each segmentation task. Optimized receptive views increase the accuracy (Chen et al., 2018a). However, the limited computational memory forces a trade-off between network depth, width, batch size, and input image size. Since deep learning makes the prediction by discretising the solution layer by layer, this has a very high computation cost. ● Secondly, the current existing U-Net models do not account for inherent noise that affects the predictions. ● Thirdly, U-Net is a discrete neural network with discrete layers, but medical image data is continuous. ● Fourthly, existing U-net variants do not provide any theoretical underpinning.

*Can one design a U-type network that overcomes the aforementioned major issues of existing models?* This is the question that we address in this work. Our work is motivated by deep implicit learning, and continuous approaches based on Neural Ordinary Differential Equations (Neural ODEs) (Chen et al., 2018b; Dupont et al., 2019) in particular. The body of literature has explored Neural ODEs mainly for image classification tasks. So far, Neural ODEs have only been used for a small number of applications as designing Neural ODEs for more complex tasks is far from being trivial.

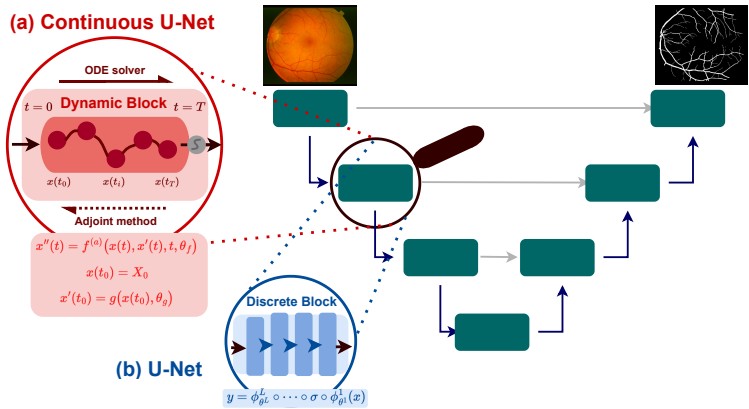

Figure 1: Visual comparison of our `continuous U-Net` vs. U-Net (and variants). The zoom-in views display the difference between discrete blocks in U-Net and our proposed dynamic blocks.

We underline that U-Nets (and majority of its variants) are designed as discrete neural networks.This is in contrast to the continuous nature of medical data. In this regard, Recurrent Neural Networks (RNNs) are more appropriate. However, non-uniform intervals for handling medical data can compromise the performance. Neural ODEs are suitable for continuous data since they output a vector field. Another major problem of the discrete neural network is the issue of overfitting/underfitting (Bilbao & Bilbao, 2017). However, we show that looking at U-Net from the lens of continuous dynamical systems, this issue can be mitigated.

Another major keypoint is that the performance of image segmentation depends partly on the receptive field of the networks. Due to the limited GPU memory, U-Net and other variants force a trade-off between architecture design and input image size. Different techniques have been developed for this problem. Examples are dilated convolutions (Folle et al., 2019) and reversible blocks (Brügger et al., 2019). These architectures reduce the computational cost because of the reduced number of stored active functions. However, the memory cost of these architectures is still associated with the depth of the model. In our work, we seek to address this problem using a trace operation via the adjoint sensitivity method with $\mathcal{O}(1)$ memory cost. That is, for neural networks based on ODEs, reversibility is built into the architecture by construction. Therefore, no matter the model's complexity, one can always provide the benefit of constant memory cost. In this work, we propose a novel family of networks that we call `continuous U-Nets` for medical image segmentation.

An overview of our approach in contrast to a standard U-Net can be found in Figure 1. In particular, our contributions are:

↻ We propose a new network, called `Continuous U-Net` , for medical image segmentation. `Continuous U-Net` is a continuous deep network whose dynamics are modelled by second order ordinary differential equations. We view the dynamics in our network as the boxes consisting of CNNs and transform them into dynamic blocks to get a solution. We introduce the first U-Net variant working explicitly in higher-order neural ODEs. We highlight:

- **Faster Convergence.** By modelling the dynamics in a higher dimension, we provide more flexibility to learn the trajectories. Therefore, `continuous U-Net` requires fewer iterations for the solution, which is more computationally efficient and in particular provides constant memory cost (Proposition 1 & Corollary 1)
- **Greater: More Robustness.** We show that `continuous U-Net` is more robust than other variants (CNNs) and provide an intuition for this (Theorem 3). Moreover, we demonstrate that our dynamic blocks are reliably useful (Theorem 1).
- **Noiseless.** We show that `continuous U-Net` is always bounded by some range whilst CNNs are not. Also, our network is smoother than existing ones leading to better handling the inherent noise in the data.
- **Underpinning Theory.** `Continuous U-Net` is the first U-type network that comes with theoretical understanding.

↻ **Open the "box" of ODE-solvers.** Existing works lack guidelines for choosing the best ODE solver. We derive qualitative measures for the choice of different ODE-solvers. At the practical level, this means that our framework can be tailor-made for various segmentation tasks (Global Error & Theorem 2).

↻ We demonstrate, through extensive numerical and visual results, that our proposed `continuous U-Net` outperforms existing U-type blocks. Moreover, we show that our proposed network stands alone performance-wise without any additional mechanism, and its performance readily competes with other mechanisms including transformers, nested U-Nets and tokenised MLP.

## 2    Related Work

The task of image segmentation, for medical data, has been widely investigated in the community, whith state-of-the-art techniques relying in U-type architectures, alone or in combination with additional mechanisms. In this section, we review the existing techniques in turn.

### 2.1    U-type Nets: A Block Based Perspective

The gold-standard U-Net model was introduced by Ronneberger et al. (Ronneberger et al., 2015) for biomedical image segmentation. U-Net consists of four main components – blocks of neural network layers, downsampling, upsampling and concatenation operations. The astonishing performance reported by U-Net motivated the fast development of a wide range of U-type variants, where the key difference is largely based on different types of blocks.

The standard U-Net architecture (Ronneberger et al., 2015) uses convolutional blocks, followed by an activation function. Further work in this area was introduced in (Zhang et al., 2018b), where the authors use residual blocks (He et al., 2016) yielding to ResUNet. To facilitate learning, the input $x$ of each convolutional block $F_\theta$ is added to the output via skip-connections so that $y = F_\theta(x) + x$. Li et al. introduced DenseUNet (Li et al., 2018) where densely connected blocks are used in the U-Net structure. Their work follows the principles of DenseNet (Huang et al., 2017). A different U-Net variant is based on inception blocks (Zhang et al., 2018a). The notion is based on computing convolutions with varying kernel sizes in parallel to then concatenate their outputs. Common kernel sizes are $1 \times 1$, $3 \times 3$ and $5 \times 5$. This approach aims at computing different levels of features by choosing different kernel sizes.

Another U-type network uses pyramid pooling blocks (Zhao et al., 2017), where the idea is to pool the input to get different input sizes (e.g., $1 \times 1 \times C$, $2 \times 2 \times C$ and $4 \times 4 \times C$ for the number of channels $C$), and apply

Table 1: Overview of properties of our `continuous U-Net` vs. existing U-type networks.

| PROPERTIES | continuous U-Net | OTHER U-NETS |
|---|:---:|:---:|
| • Parameters Efficiency | ✓ | ✗ |
| • Constant Memory Cost | ✓ | ✗ |
| • Continuous Network | ✓ | ✗ |
| • Noise Resistant | ✓ | ✗ |
| • Beyond homomorphic Transformations | ✓ | ✗ |
| • Reversible | ✓ | ✗ |
| • Theoretical Underpinning | ✓ | ✗ |

convolutions on each of them before upsampling to the original input size. The upsampled output is then concatenated channel-wise. Most recently, the work of Pinckaers et al. (Pinckaers & Litjens, 2019) uses first order neural ODEs in a U-Net setting.

## 2.2 U-type Nets with Additional Mechanisms

The aforementioned techniques seek to create variants of U-Net solely based on the block types. Another set of techniques has instead focused on creating additional mechanism in U-Net structures. Zhang et al. introduced ResUNet (Zhang et al., 2018b), where residual blocks along with several additional mechanisms such as atrous convolutions (Chen et al., 2017), pyramid scene parse pooling (Zhao et al., 2017) and multi-task inference (Ruder, 2017) were used. An attention mechanism was introduced in U-Net where the combination of both is known as Attention U-Net (Oktay et al., 2018). Attention gates are used to filter features before concatenating the upsampled input and the skip connection in the decoder part of the U-Net.

In more recent works, DynUNet (Ranzini et al., 2021) was introduced as a combination of two works. It takes the heuristic rules and setting from nnU-Net (Isensee et al., 2019), and the optimisation scheme of Futrega et al. (Futrega et al., 2022) for searching an (sub-)optimal network structure. Self-attention mechanisms have also been explored for U-Net. Chen et al. proposed TransUNet (Chen et al., 2021) that combines a transformer encoder, with a self-attention mechanism, and a classical convolutional neural network decoder. The most recent U-type network, called UNeXt, was introduced in (Valanarasu & Patel, 2022). UNeXt is a convolutional multi-layer perceptron (MLP) based network, which uses tokenised MLP blocks with axial shifts. In comparison to transformer based approaches like TransUNet, UNeXt needs only a small number of parameters.

## 2.3 Existing Techniques & Comparison to Ours

We provide a summary of properties for `continuous U-Net` and existing U-Type nets in Table 1. More precisely, existing U-type networks (Ronneberger et al., 2015; Zhang et al., 2018b; Li et al., 2018; Zhang et al., 2018b; Oktay et al., 2018; Ranzini et al., 2021; Valanarasu & Patel, 2022) discretise the solution layer by layer which yields to high computational cost. Contrary, our work is a continuous architecture that can be solved via the adjoint method(Chen et al., 2018b), which translates in constant memory cost. Unlike existing U-type blocks (Ronneberger et al., 2015; Zhang et al., 2018b; He et al., 2016; Li et al., 2018; Zhang et al., 2018a; Zhao et al., 2017), we propose new dynamic blocks modelled by second-order neural ODEs, which are not restricted to homomorphic transformations. Moreover, opposite to (Pinckaers & Litjens, 2019) our dynamic blocks are at least twice continuously differentiable resulting in being robust to noise.

Most recently, new mechanisms have been used along with U-Net including attention, self-attention, transformers, heuristic rules and nested U-Nets (Zhang et al., 2018b; Oktay et al., 2018; Ranzini et al., 2021; Chen et al., 2021; Valanarasu & Patel, 2022). Unlike these works, our `continuous U-Net` does not use any additional mechanism. We instead introduce new dynamic blocks. Therefore, the philosophy of our work is closer to the block based U-Net techniques (Ronneberger et al., 2015; Zhang et al., 2018b; He et al., 2016; Li et al., 2018; Zhang et al., 2018a; Zhao et al., 2017) and not directly comparable to those with additional

mechanisms. We highlight that our `continuous U-Net` does not use any additional mechanism, *opening the door to a new research line on continuous U-type networks.* Finally and to the best of our knowledge, this is the first U-type architecture that provides underpinning theory.

## 3 Proposed Technique

This section contains the key parts of our proposed `continuous U-Net` : (i) we present our proposed dynamic blocks and how we model the dynamics of our network using second order Neural ODEs, (ii) our derived quality measure for tailor-made segmentation tasks and (iii) robustness and noise properties of our network.

### 3.1 Unboxing Continuous U-net

`Continuous U-Net` greatly differs from existing U-type networks since it is modelled as a continuous approach. That is, we avoid computing predictions by discretising the solution layer-by-layer, which involves a high computational cost. In particular, our network models the dynamics by taking the CNN boxes and transforming them into ODE blocks. Unlike existing works on Neural ODEs, we go beyond the standard learning setting by designing a new U-type architecture using higher order neural ODEs. *Why designing higher-order blocks?* The standard Neural ODE setting fails to learn complex flows. As the training progresses and the flow becomes more complex, the number of steps required to solve the ODE increases (Chen et al., 2018b; Grathwohl et al., 2018). This is one of the limitation of neural ODEs. Although augmented neural ODEs (ANODE) (Dupont et al., 2019) were proposed to mitigate this issue to some degree, ANODE is still a first order ODE. This property of first order ODE limits the performance in terms of computational speed and learning of the flow. Moreover and unlike our work, those approaches are stand-alone techniques whilst our work uses a different principle – building on ODE blocks to construct a new U-type network that can handle segmentation for complex data as in the medical domain.

**Dynamic Blocks.** `Continuous U-Net` is formulated from a dynamical systems perspective. We transform the CNN boxes into second order ODEs blocks. Our blocks work under the definition of second order Neural ODEs, which read:

$$\begin{cases} x''(t) = f^{(a)}\left(x(t), x'(t), t, \theta_f\right) \\ x(t_0) = X_0, \quad x'(t_0) = g(x(t_0), \theta_g), \end{cases} \tag{1}$$

whose velocity is described by a neural network $f^{(a)}$ with parameters $\theta_f$ and initial position given by the points of a dataset $X_0$. We now discuss how our dynamic block can be reliably useful and computational efficient.

> **Proposition 1**
>
> Any given high-order Neural ODEs can be transformed into a system of first-order Neural ODEs.

*Proof.* Consider a first order Neural ODEs:

$$\begin{cases} x'(t) = f^{(v)}(x(t), t, \theta_f), \\ x(t_0) = X_0 \end{cases} \tag{2}$$

where velocity is described by a neural network $f^{(v)}$ with parameters $\theta_f$. Define an $m^{th}$ order neural ODEs as:

$$\begin{cases} x^m(t) = f^{(a)}(x(t), x'(t), ..., t, \theta_f), \\ x(t_0) = X_0, \\ \vdots \\ x^{m-1}(t_0) = g(x(t_0), x'(t_0), ..., \theta_g) \end{cases} \tag{3}$$

where $m \in \mathbb{Z}^+$, and $f^{(a)}$ refers to a neural network with parameters $\theta_f$.

$$Let \ \mathbf{z}(t) = \begin{bmatrix} x(t) = x_1(t) \\ x'(t) = x_2(t) \\ \vdots \\ x^{(m-1)}(t) = x_m(t) \end{bmatrix}, \mathbf{z}(t_0) = \begin{bmatrix} x(t_0) \\ x'(t_0) \\ \vdots \\ x^{m-1}(t_0) \end{bmatrix} \tag{4}$$

then the $m^{th}$ order Neural ODEs becomes:

$$\mathbf{z}'(t) = \begin{bmatrix} x_2(t) \\ x_3(t) \\ \vdots \\ f^{(a)}(x_1(t), ..., x_m(t), t, \theta_f) \end{bmatrix} = f^{(v)}(\mathbf{z}(t), t, \theta_f) \tag{5}$$

We therefore can always transform an $m^{th}$ order Neural ODEs into a first order Neural ODEs. Note that $f^{(a)}$ and $f^{(v)}$ denote networks with distinct weight sets. $\qquad\square$

The way to demonstrate that the $m^{th}$ order ODE is reliably useful is through well-posedness.

> **Theorem 1**
>
> An $m^{th}$ Order Neural ODEs with neural network $f^{(v)}$ is well-posed (in the sense of Hadamard) if $f^{(v)}$ is Lipschitz.

*Proof.* Using Proposition 1, we show that any given high order ODE can be transformed into a system of first order ODEs. Since neural network is usually Lipschitz (e.g.,(Virmaux & Scaman, 2018; Fazlyab et al., 2019)) so we can sssumed $f^{(v)}$ is lipschitz. Let f : $[0 , T] \times \mathbb{R}^d \to \mathbb{R}^d$ be a continuous function depends on t and uniformly Lipschitz in x . Let $X_0 \in \mathbb{R}^d$ , by Picard's Existence Theorem , Then there exists a unique differentiable x : $[0,T] \to \mathbb{R}^d$ satisfying

$$\begin{cases} x'(t) = f^{(v)}(x(t), t, \theta_f), \\ x(t_0) = X_0 \end{cases} \tag{6}$$

It is then to say that if

`Continuous U-Net` breaks the barrier of the receptive field limitation that existing U-type networks have. In existing networks, the key factor affecting the memory cost is the storing of intermediate hidden unit activation functions. Our dynamic blocks only need a single point to reconstruct the entire trajectory by forward and backward iterations. That is, `continuous U-Net` offers constant memory cost for segmentation. We guarantee this property as our blocks use trace operation via the adjoint sensitivity method with $\mathcal{O}(1)$ memory cost.

> **Corollary 1**
>
> Our second order Neural ODE's dynamic blocks can be solved by using the first order adjoint method.

*Proof.* Our dynamic blocks are based on second order neural ODEs. We therefore set the order to $m = 2$ in Proposition 1 which implies $\mathbf{z}'(t) = f^{(v)}(\mathbf{z}(t), t, \theta_f)$. $\qquad\square$

Our second order dynamic blocks can then be solved by first-order adjoint method. At the practical level, this means that we do not need to store the layers one-by-one in the architecture. We need any single point to reconstruct the entire trajectory by forward and backward iterations. Therefore, we offer a constant memory cost.

### 3.2 Opening the ODE-solver Box

When working with Neural ODEs an open question is – how to choose the best ODE solver? In this section, we derive qualitative measures for the choice of different ODE-solvers. At the implementation level, this means that our framework can be tailor-made for various segmentation tasks. In this section, we open the ODE-solver 'box' and derive an error analysis for better understanding on how to choose different solver for `continuous U-Net` .

> **Global Error**
>
> Global Error (GE): $e_n = x(t_n) - x_n$ , where $x(t_n)$ denotes the exact solution and $x_n$ denotes the numerical solution. GE is the error at final time.

**Property 1** ($e_n \propto h$)**.** *If $h$ is small, higher accuracy results can be achieved. Moreover, if $z = \mathcal{O}(h^p)$, we say $z$ is p-th order where $|z| \leq Ch^p$ , $C > 0$ for all $0 < h < h_0$. That is, when $p$ is larger, the numerical method converges faster, being a better method in terms of convergence rate. In order to be convergent, we need to satisfy the conditions of stability and consistency.*

*How GE can be used?* Euler's method is the most simple yet widely used one. We take it as a good starting point for more advanced methods and use our defined GE to provide further intuition.

> **Lemma**
>
> For any real number $x \geq 0$ , then the exponential function $e^x$ is always $\geq 1 + x$ .

*Proof.* Let $f(x) = e^x - 1 - x$ , then $f(0) = 1 - 1 = 0$. Moreover, $f'(x) = e^x - 1$ but $e^x$ always $\geq 1$ implies $f'(x)$ is positive and $f(x)$ is increasing function. Thus, $f(x) \geq f(0)$. We get $f(x) = e^x - 1 - x \geq 0$ implies $e^x \geq 1 + x, \forall x \geq 0$ □

> **Theorem 2**
>
> Euler's method in our dynamic blocks (based on neural ODEs) converges
>
> $$\begin{cases} x'(t) = \lambda x(t) + f^{(v)}(t, \theta_f) & 0 \leq t \leq t_f \\ x(0) = 1 & \lambda \in \mathbb{C} \end{cases} \tag{7}$$
>
> and the GE at t $\in [0, t_f]$ is $\mathcal{O}(h)$.

*Proof.* We follow the idea from (Griffiths & Higham, 2010) and extend to Neural ODEs. Euler's method for our dynamic blocks reads

$$x_{n+1} = x_n + \lambda h x_n + h f^{(v)}(t_n, \theta_f) = (1 + \lambda h) x_n + h f^{(v)}(t_n, \theta_f), \tag{8}$$

applying a Taylor expansion to the exact solution, we get

$$x(t_{n+1}) = x(t_n) + h(\lambda x(t_n) + f^{(v)}(t_n, \theta_f)) + R_1(t_n). \tag{9}$$

We can get the formula of GE from the difference of equation 8 and equation 9:

$$\begin{aligned} e_{n+1} &= x(t_{n+1}) - x_{n+1} \\ &= (1 + \lambda h)(x(t_n) - x_n) + T_{n+1} = (1 + \lambda h)e_n + T_{n+1} \end{aligned} \tag{10}$$

We know that $e_0 = 0$. The relation of GE for different step is given in equation 10. We can combine this with the local truncation error. Substituting $n = 0, 1, 2$ into equation 9 and using $e_0 = 0$, $e_1 = T_1$ and

$e_2 = (1 + h\lambda)e_1 + T_2 = (1 + h\lambda)T_1 + T_2$ gives

$$e_n = (1 + h\lambda)^{(n-1)}T_1 + (1 + h\lambda)^{n-2} + ... + T_n$$
$$= \sum_{j=1}^{n}(1 + h\lambda)^{n-j}T_j \tag{11}$$

By Lemma 1 , we know $e^{h|\lambda|} \geq 1 + h|\lambda| \geq |1 + h\lambda|$. Similarly, $e^{|\lambda|t_f} \geq e^{(n-j)h|\lambda|} = e^{|\lambda|t_{n-j}} \geq |1 + h\lambda|^{n-j}$. We know that $|T_j| \leq Ch^2$ which implies,

$$e_n = \sum_{n=1}^{j}(1 + \lambda h)^{n-j}T_j \leq ne^{|\lambda|t_f}ch^2 \tag{12}$$

Since $nh$ is finite which implies $e_n$ is also finite. Thus, $|e_n| \leq Ch$ , $e_n = \mathcal{O}(h)$ Therefore, Euler's method applied to our dynamic blocks converges at a first-order rate, i.e., $p = 1$. □

Our Theorem 2 focuses on the Euler method, which is a simple RK method with order 1. We would like to clarify that since higher-order RK methods are extensions of Euler's method, proving the theorem for Euler's method establishes a baseline from which the behavior of higher-order methods can be inferred. We remark that this foundational proof is sufficient to demonstrate the theoretical properties we are showcasing, as higher-order RK methods would follow the same conceptual line, with additional complexity that does not change the underlying theory presented.

Whilst Euler's method is widely used as it is simple and always convergent, it is not accurate, and limited at the practical level. The GE is only $\mathcal{O}(h)$. We therefore provide further intuition on the family of linear multistep method (LMMs). LMMs provide a higher order $p$ which converges faster than Euler's method.

For example, explicit Adams-Bashforth and implicit Adams-Bashforth-Moulton are LMMs with a higher convergent rate. The GE of these methods is $\mathcal{O}(h^2)$. From a mathematical point of view, it is known that a convergent LMM is consistent. However, the reverse does not hold, and *Zero-stability*[1] is also required (Dahlquist, 1956). That means that Consistent + zero-stability imply convergence for these models. If we want a method to be convergent, we need both conditions. This is the intuition behind why high order in linear multistep methods is not possible. Finally, we discuss the Runge–Kutta (RK) method, which is another widely used and effective method. Euler's method and midterm method are also type of RK methods with lower order. RK methods are one-step methods composed of a number of stages. The midpoint method is a RK2, i.e., a two stage method while Euler's method is RK1, i.e., a 1 stage method. *Is it possible to always find a RK method of s stages (order s)?* This is not true for $s > 4$.

*What method is the best then?* RK methods are a one-step methods, and the linear multistep method is a two step method. Compared to linear multistep, RK does not have to treat the first few steps taken by a single-step integration method as special cases. Moreover, RK methods are very stable. Based on our derived error analysis along with the aforementioned factors, we found that our continuous blocks, `continuous U-Net` for segmentation, greatly benefit performance-wise when using RK4. Our analysis in this section provides an answer to the open question on the solver when working with Neural ODEs. We remark that existing works on neural ODEs lack such analysis to provide a clear guidance on the solver.

### 3.3 Continuous U-net: Greater and Noiseless

This section provides the intuition on why our `continuous U-Net` is more robust and noiseless than existing U-type architectures.

Our dynamic blocks are defined as second order Neural ODEs. Existing works have shown empirically that the properties inherent to ODEs make them more robust than CNNs in terms of robustness, e.g., (Yan et al., 2019). The key idea to show such robustness comes from the ODEs integral curves as follows, which is detailed next.

---

[1]Stability, specifically zero stability, relates to the method's capability to manage error propagation throughout the computational iterations. This is assessed by the root condition: all roots of the characteristic polynomial should satisfy $|r| \leq 1$, with roots where $|r| = 1$ being simple. Consistency involves the method's local truncation error reducing to zero as the step size $h$ approaches zero.

---

**Theorem 3**

ODE integral curves do not intersect (Coddington & Levinson, 1955, Ch. 1). If $z_1(t)$ and $z_2(t)$ are two ODE solution with different initial value of the same function, then $z_1(t) \neq z_2(t)$ for all $t \in [0, \infty)$. The proof follows (Coddington & Levinson, 1955).

---

*Why is* Theorem 3 *relevant?* Since ODE integral curves do not intersect, we can prove that some range always bounds the output of our dynamic blocks whilst the output of CNNs is not bounded by any range. This proof leads to Neural ODEs being more robust than CNNs. In particular, for our dynamic blocks based on second order ODEs, we can further support the robustness as follows: If we take a derivative on a function, we require the function to be smooth. A significant difference between the first-order ODE and second-order ODE is that the latter is at least twice continuously differentiable. However, a first-order ODE requires once continuously differentiability yielding to our dynamic blocks being smoother than first-order ODEs – and therefore less sensitive to noise than existing techniques. That is, the inherent smoothness of second-order ODEs, characterised by at least twice continuously differentiable solutions, contributes to noise suppression by ensuring a smoother trajectory of the dynamic block's output. This smoother trajectory is less affected by small variations in initial conditions, which translates to lower sensitivity to input noise.

Secondly, existing U-type networks can only learn smooth homeomorphisms, which is one of the modelling disadvantages. Our blocks solve this problem by providing extra dimensions through the second order design. Moreover, our dynamic blocks are a physics-based model that better captures the nature of segmentation. We also remark that our second-order dynamic blocks are parameter efficient as no parameters are required on $x'(t_0) = g(x(t_0), \theta_g)$ That's why second order Neural ODEs is most robust compared to first order Neural ODEs.

*Why our* `continuous U-Net` *improves for segmentation?* Our model integrates second-order ODE dynamics, which inherently allows for variable receptive field sizes without the need for manual optimisation. This adaptability leads to enhanced accuracy while circumventing the computational trade-offs imposed by limited memory, which traditional U-Nets cannot avoid. The continuous nature of our model aligns more closely with the intrinsic continuity of medical image data, leading to more natural and accurate segmentation outputs. Moreover, the built-in reversibility of our architecture, enabled by the adjoint sensitivity method, maintains constant memory cost regardless of model complexity. This contrasts sharply with discrete U-Nets that lack theoretical underpinning and struggle with memory efficiency.

Moreover, we remark that our second-order neural ODE model inherently resists noise due to its smoother numerical solutions, which arise from the model's twice continuously differentiable nature. This smoothness is a structural property of the model itself, rather than a result of training on varied data. While data augmentation can indeed increase robustness by introducing more variance during training, it does not inherently smooth the function that the network learns. Instead, it makes the model more tolerant to the types of variations seen during training.

## 4 Experiments

In this section, we detail the experiments that we conducted to validate our `continuous U-Net` model.

### 4.1 Data Description & Evaluation Protocol

We expensively evaluate our `continuous U-Net` using six medical imaging datasets. They are highly heterogenous covering a wide range of medical data and significantly varying in terms of image sizes, fidelity of segmentation masks and dataset sizes. An overview of the datasets used and their properties can be found in Table 2.

**Evaluation Protocol.** Following a standard protocol in medical image segmentation, we evaluate the performance of the proposed `continuous U-Net` and existing techniques using three metrics: the Dice score, accuracy and averaged Hausdorff distance. For a fair comparison, we use a shared code-base for all experiments. More precisely, we set a learning rate of $1 \times 10^{-3}$, a step-based learning rate scheduler with a

Table 2: Characteristics of the datasets used in our experiments.

| Dataset | # Samples | # Train | # Test | Image size |
|---|---|---|---|---|
| GlaS Challenge (Sirinukunwattana et al., 2017) | 165 | 85 | 80 | 352x352 |
| STARE (Hoover et al., 2000) | 20 | 16 | 4 | 512x512 |
| Kvasir-SEG (Jha et al., 2020) | 1000 | 800 | 200 | 256x256 |
| Data Science Bowl (Caicedo et al., 2019) | 841 | 707 | 134 | 256x256 |
| ISIC Challenge (Gutman et al., 2016) | 1279 | 900 | 379 | 512x512 |
| Breast Ultrasound Images (Al-Dhabyani et al., 2020) | 647 | 518 | 129 | 256x256 |

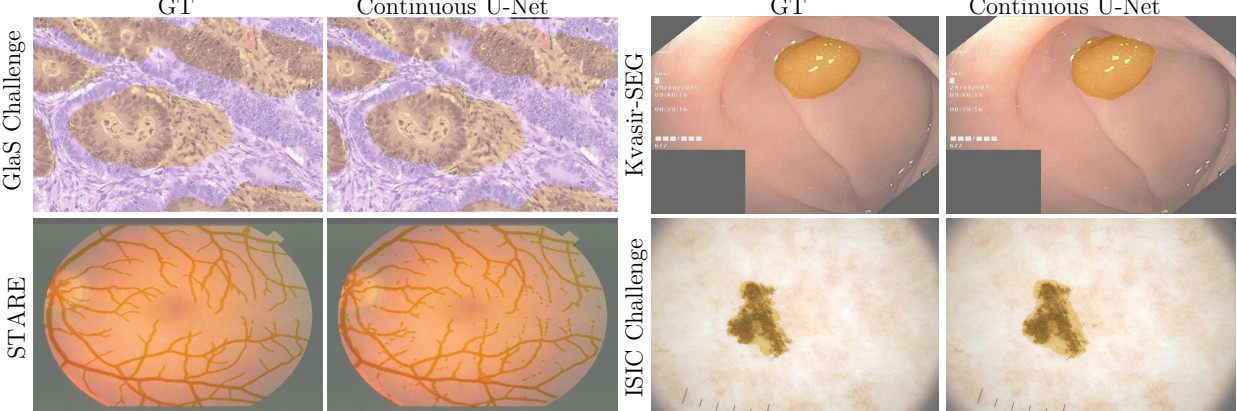

Figure 2: Visual segmentation results from our proposed continuous U-Net, with the output masks overlaid in yellow for improved contrast and comparison. The left column shows the Ground Truth (GT) images, while the right column presents the corresponding segmentation by the continuous U-Net. Extended results and comparisons against all techniques are available in the supplementary material.

Table 3: Comparison in terms of Dice score, accuracy and average Hausdorff distance for existing U-Net blocks and our dynamic blocks. We denote our `continuous U-Net` with ★. The best results are highlighted in green colour.

| | Block | GlaS Challenge | | | STARE Dataset | | | Kvasir-SEG Dataset | | | DS Bowl Dataset | | |
|---|---|---|---|---|---|---|---|---|---|---|---|---|---|
| | Type | Dice↑ | Acc↑ | AHD↓ | Dice↑ | Acc↑ | AHD↓ | Dice↑ | Acc↑ | AHD↓ | Dice↑ | Acc↑ | AHD↓ |
| U-Net | PLN | 0.7616 | 0.7824 | 13.27 | 0.7828 | 0.9462 | 8.79 | 0.4597 | 0.8595 | 3.96 | 0.9156 | 0.9681 | 3.68 |
| | RSE | 0.7893 | 0.8100 | 12.15 | 0.7731 | 0.9460 | 8.74 | 0.6418 | 0.8754 | 4.57 | 0.9131 | 0.9687 | 3.67 |
| | DSE | 0.7264 | 0.7516 | 13.93 | 0.7358 | 0.9413 | 8.67 | 0.6071 | 0.8649 | 4.58 | 0.9181 | 0.9690 | 3.63 |
| | INC | 0.7962 | 0.8192 | 11.58 | 0.7750 | 0.9505 | 8.33 | 0.6856 | 0.8825 | 4.30 | 0.9210 | 0.9724 | 3.41 |
| | FO | 0.8037 | 0.8270 | 10.47 | 0.5005 | 0.9241 | 6.28 | 0.7465 | 0.9080 | 3.58 | 0.8953 | 0.9625 | 4.00 |
| | PSP | 0.5523 | 0.5699 | 14.37 | 0.4805 | 0.9251 | 5.77 | 0.5883 | 0.8672 | 4.36 | 0.9060 | 0.9655 | 3.74 |
| ★ | DB | 0.8469 | 0.8675 | 9.31 | 0.8378 | 0.9568 | 8.21 | 0.7922 | 0.9243 | 3.32 | 0.9335 | 0.9745 | 3.29 |

| | Block | BUSI Dataset | | | ISIC Challenge | | |
|---|---|---|---|---|---|---|---|
| | Type | Dice↑ | Acc↑ | AHD↓ | Dice ↑ | Acc↑ | AHD↓ |
| U-Net | PLN | 0.4724 | 0.9012 | 2.56 | 0.8232 | 0.8764 | 6.76 |
| | RSE | 0.7179 | 0.9218 | 2.63 | 0.8501 | 0.9003 | 6.00 |
| | DSE | 0.7095 | 0.9196 | 2.61 | 0.8376 | 0.8907 | 6.32 |
| | INC | 0.7434 | 0.9243 | 2.71 | 0.8570 | 0.9050 | 5.83 |
| | FO | 0.7893 | 0.9338 | 2.44 | 0.9087 | 0.9452 | 4.45 |
| | PSP | 0.6098 | 0.9197 | 2.46 | 0.8179 | 0.8798 | 6.67 |
| ★ | DB | 0.8090 | 0.9447 | 2.15 | 0.9094 | 0.9433 | 4.46 |

step size of 1 and a gamma value of 0.999. We use a fourth-order Runge-Kutta (RK4) solver, a batch size of 16 and train all networks for 500 epochs.

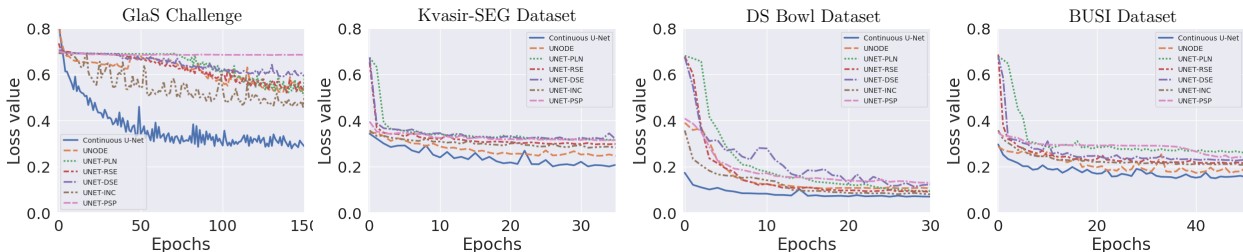

Figure 3: Convergence plots for different types of U-Net blocks. The behaviour of our dynamic blocks is denoted in blue colour, where only a few iterations are needed to converge whilst existing blocks need far longer iterations.

Table 4: Performance comparison, in terms of Dice score, on a range of noise level We denote our `continuous U-Net` with ★. The best results are highlighted in green colour. Refer to the supplementary material for extended results.

| | | NOISE LEVEL EXPERIMENTS | | | | | | | | | | | |
|---|---|---|---|---|---|---|---|---|---|---|---|---|---|
| | Block | BUSI Dataset | | | | Kvasir-Seg Dataset | | | | ISIC Challenge | | | |
| | Type | 0.0 | 0.2 | 0.4 | 0.5 | 0.0 | 0.2 | 0.4 | 0.5 | 0.0 | 0.2 | 0.4 | 0.5 |
| U-Net | PLN | 0.4724 | 0.4723 | 0.4722 | 0.4721 | 0.4597 | 0.4597 | 0.4597 | 0.4597 | 0.8232 | 0.4064 | 0.4064 | 0.4064 |
| | RSE | 0.7179 | 0.2373 | 0.2130 | 0.2094 | 0.6418 | 0.4597 | 0.4597 | 0.4597 | 0.8501 | 0.4064 | 0.4064 | 0.4064 |
| | DSE | 0.7095 | 0.1944 | 0.2321 | 0.2423 | 0.6071 | 0.4597 | 0.4597 | 0.4597 | 0.8376 | 0.4774 | 0.4067 | 0.4064 |
| | INC | 0.7434 | 0.1930 | 0.1769 | 0.1883 | 0.6856 | 0.4597 | 0.4597 | 0.4597 | 0.8570 | 0.4064 | 0.4064 | 0.4064 |
| | PSP | 0.6098 | 0.5899 | 0.5368 | 0.5255 | 0.5883 | 0.4668 | 0.4597 | 0.4597 | 0.8179 | 0.3979 | 0.3851 | 0.3847 |
| | FO | 0.7754 | 0.5345 | 0.5154 | 0.5131 | 0.7224 | 0.6401 | 0.5964 | 0.5915 | 0.9031 | 0.6941 | 0.6754 | 0.6612 |
| ★ | DB | 0.8090 | 0.6881 | 0.5743 | 0.5569 | 0.7922 | 0.6417 | 0.6164 | 0.5972 | 0.9094 | 0.7055 | 0.6910 | 0.6746 |

Table 5: Ablation study on different types of ODE solvers on four medical image segmentation datasets. Fourth-order Runge-Kutta (RK4) consistently outperforms Euler, Explicit Adams-Bashforth (AB) and Implicit Adams-Bashforth-Moulton (ABM) in Dice score, accuracy and average Hausdorff distance. The best results are highligthed in green colour. Refer to the supplementary material for extended results.

| Solver | STARE Dataset | | | DS Bowl Dataset | | | Kvasir-SEG Dataset | | | ISIC Challenge | | |
|---|---|---|---|---|---|---|---|---|---|---|---|---|
| Type | Dice↑ | Acc↑ | AHD↓ | Dice ↑ | Acc↑ | AHD↓ | Dice ↑ | Acc↑ | AHD↓ | Dice ↑ | Acc↑ | AHD↓ |
| Euler | 0.7697 | 0.9468 | 8.57 | 0.8911 | 0.9539 | 4.37 | 0.5450 | 0.8596 | 4.82 | 0.8487 | 0.9003 | 6.91 |
| AB | 0.8066 | 0.9517 | 8.49 | 0.9252 | 0.9726 | 3.39 | 0.7850 | 0.9185 | 3.46 | 0.9051 | 0.9398 | 4.68 |
| ABM | 0.8015 | 0.9506 | 8.50 | 0.9246 | 0.9730 | 3.37 | 0.7714 | 0.9158 | 3.50 | 0.9032 | 0.9374 | 4.78 |
| RK4 | 0.8378 | 0.9568 | 8.2149 | 0.9335 | 0.9745 | 3.29 | 0.7922 | 0.9243 | 3.32 | 0.9094 | 0.9433 | 4.46 |

## 4.2 Results & Discussion

In this section, we present all experimental results and visualisation conducted to validate our approach.

● **Comparison to Other Existing U-Type Blocks.** We underline that our work is a stand-alone continuous network, where the highlight is the new dynamic block. Therefore, our main comparison is based on different types of U-Net blocks. More precisely, our comparison includes plain convolutional blocks (PLN), residual blocks (RSE), dense blocks (DSE), inception blocks (INC), pyramid pooling blocks (PSP), first order ODE blocks (FO) and our dynamic blocks (DB). We start by reporting the global results in Table 3. The displayed numbers show all metrics. Looking at the results more closely, we observe that we achieve significant improvement over most SOTA techniques, most notable on the GlaS and the Kvasir-SEG challenge datasets. There are only three cases where the FO block performs similarly well than our dynamic blocks. However, a closer look at the FO block shows general difficulties to get reliable performance for the remaining datasets – as for example in the STARE dataset. In contrast, our dynamic blocks report a stable and high performance across all datasets and metrics. Additionally, Figure 2 highlights visual results for our approach on different datasets.

Table 6: Performance comparison of `continuous U-Net` against state-of-the-art U-Net architectures with additional mechanisms. We denote our `continuous U-Net` with ★. Extended results with additional three datasets can be found in the supplementary material.

| TECHNIQUE | MECHANISM | GlaS Challenge | | | STARE Dataset | | | Kvasir-SEG Dataset | | |
|---|---|---|---|---|---|---|---|---|---|---|
| | | Dice↑ | Acc↑ | AHD↓ | Dice↑ | Acc↑ | AHD↓ | Dice↑ | Acc↑ | AHD↓ |
| Attn. U-Net | Attn. Gates | 0.7991 | 0.8173 | 12.10 | 0.8675 | 0.9637 | 7.63 | 0.7290 | 0.8997 | 4.35 |
| DynU-Net | Heuristic-R + Opt | 0.8626 | 0.8828 | 9.15 | 0.8508 | 0.9575 | 8.25 | 0.7634 | 0.9101 | 4.00 |
| U2Net | Nested U-Nets | 0.8465 | 0.8658 | 9.18 | 0.7845 | 0.9539 | 7.84 | 0.7950 | 0.9171 | 3.23 |
| UNeXt | Tokenised MLP | 0.8911 | 0.9040 | 7.72 | 0.7866 | 0.9482 | 8.59 | 0.7879 | 0.9260 | 3.08 |
| TransUnet | Transformers | 0.8984 | 0.9145 | 7.11 | 0.8565 | 0.9607 | 7.79 | 0.8749 | 0.9429 | 2.70 |
| ★ | ✗ | 0.8469 | 0.8675 | 9.31 | 0.8378 | 0.9568 | 8.21 | 0.7922 | 0.9243 | 3.32 |

● `Continuous U-Net` : **Faster & Greater** To further support our theoretical findings, we evaluate the convergence of our `continuous U-Net` against the blocks of other existing techniques. In Figure 3 we display convergence plots for challenging datasets. We observe that our theory agrees with the experiments; as our dynamic blocks offer an extra dimension, by the second-order Neural ODEs modelling, which yields to require fewer iterations for the solution than other existing U-type blocks. In a closer look, we observe that our dynamic blocks only need a small number of epochs to converge whilst the other ones do not converge even with a higher number of epochs.

● `Continuous U-Net` : **Noiseless** In Subsection 3.3, we proved that our `continuous U-Net` is less sensitive to noise than other CNN-based U-Net architectures. We use pretrained models, adding zero mean Gaussian noise with varying standard deviations $(0.2, 0.4, 0.5)$ and compute the Dice score for each of them. Table 4 reports the results and highlights the robustness of our approach. `Continuous U-Net` consistently achieves the highest Dice scores for all noise levels. Additionally, comparing our `continuous U-Net` approach to the second best performing version – the U-Net with inception blocks (INC) – the difference becomes even more obvious. For all datasets, adding even a small amount of noise (standard deviation of 0.2) to the INC model, the Dice score drops massively, e.g., from 0.7434 to 0.1930 on the GlaS challenge dataset. Our approach, however, is able to deal much better with additive noise, leading to a difference in Dice score of only 0.1209. A similar pattern can be observed for the other two datasets.

● `Continuous U-Net` : **Opening the ODE-Solver Box** Subsection 3.2 sheds light onto the theoretical error analysis for different types of ODE solvers. We compare Euler's method, Explicit Adams-Bashforth (AB), Implicit Adams-Bashforth-Moulton (ABM) and a fourth-order Runge-Kutta (RK4) Table 5 reports the performance for all methods and datasets. Our derived qualitative measures agree with the empirical results, the Runge-Kutta method outperforms Euler, Adam-Bashforth and Adam-Bashforth-Moulton for each of the datasets in all metrics and is thus suggested for segmentation tasks.

● `Continuous U-Net` **vs. other U-type with Additional Mechanisms.** We underline that our main contribution in this paper is to create a continuous network with our dynamic blocks. Whilst a direct comparison with architectures using additional mechanisms is unfair, we ran a set of experiments on current SOTA techniques with additional mechanisms including attention gates, transformers or tokenised MLPs. Firstly, we seek to demonstrate that our network stands alone without any additional mechanism. Secondly, to open the door to a new research line to design continuous U-type networks with additional mechanisms– which is far from being trivial. Table 6 provides the results for these experiments. `Continuous U-Net` is able to outperform at least two methods per dataset (GlaS challenge, STARE dataset, DS Bowl dataset, ISIC challenge), three for the Kvasir-SEG dataset and even four on the BUSI dataset.

We also compare our network's efficiency with SoTA models, emphasising our unique achievement in computational efficiency without relying on complex enhancements. Specifically, our model's performance on the ISIC dataset stands out, requiring only 0.45 GFLOPs per inference, which is notably lower than UNeXt's 0.57 GFLOPs and TransUnet's 38.52 GFLOPs. This efficiency leads to a significant reduction in parameters and model size, alongside a boost in inference speed. Our model outperforms the most efficient discrete U-Nets, including UNeXt, across several benchmarks, without the need for tokenised MLPs or other such mechanisms, highlighting our model's ability to guarantee constant memory cost, an advantageous trait in deep learning design.

# 5 Conclusion

`Continuous U-Net` is a continuous network modelled by our dynamic blocks using second order neural ODEs. We show that our approach outperforms existing U-Net blocks on six benchmarking datasets, and readily competes or even outperforms famous U-Net architectures with additional mechanisms like attention. By parameterising the model as a continuous function over time, we offer a novel perspective that allows for repeated computations within the same 'layer'—a conceptual layer defined by the continuous time variable $t$. This is in contrast to traditional U-Nets that perform a single computation per discrete layer. Hence, our approach can be seen as improving the capabilities of U-Nets. Our findings open the door to a new research line on continuous U-type networks by introducing the to the best of our knowledge first U-type architecture that provides underpinning theory. Future work includes a theoretical extension on higher-order ODEs and integrating additional mechanisms like attention.

# 6 Acknowledgements

CWC acknowledges support from Department of Mathematics, College of Science , CityU, HKASR reaching out award and funding from CCMI, University of Cambridge. RHC acknowledges support from HKRGC GRF grants CityU1101120 and CityU11309922 and CRF grant C1013-21GF. AAR gratefully acknowledges funding from the Cambridge Centre for Data-Driven Discovery and Accelerate Programme for Scientific Discovery, made possible by a donation from Schmidt Futures, ESPRC Digital Core Capability Award, and CMIH, CCMI, University of Cambridge. CBS acknowledges support from the Philip Leverhulme Prize, the Royal Society Wolfson Fellowship, the EPSRC advanced career fellowship EP/V029428/1, EPSRC grants EP/S026045/1 and EP/T003553/1, EP/N014588/1, EP/T017961/1, the Wellcome Innovator Awards 215733/Z/19/Z and 221633/Z/20/Z, the European Union Horizon 2020 research and innovation programme under the Marie Skodowska-Curie grant agreement No. 777826 NoMADS, the Cantab Capital Institute for the Mathematics of Information and the Alan Turing Institute.

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

# A   Appendix

This document extends the practicalities and results presented in the main paper. This is structured as follows.

- **Supplementary Convergence & Noise Results:** We provide further results comparisons of our `continuous U-Net` against other existing U-type networks. We also give further experimental results on noisy data.

- **Supplementary Solver Results:** We provide further results on the ablation study of our model under different type of ODE solvers and its effect.

- **Supplementary Performance Comparison & Training Scheme.** We give further numerical comparisons for different type of blocks along with other type of U-Nets with different mechanisms. In the interest of clarity and completness, we give an explicit definition in the training setting of our model.

## A.1   Supplementary Convergence & Noise Results

Figure 4 provides convergence plots for different types of U-Net blocks for the STARE dataset and the ISIC challenge. Our `continuous U-Net` is denoted in blue colour. In a closer look, we observe that our model only need a few iterations to converge whilst existing blocks need far more iterations.

We also extend our noise results from the main paper using the GlaS challenge, STARE dataset and DS Bowl dataset. Our experients are conducted by adding zero mean Gaussian noise with increasing standard deviation. The results are in Table 8. They shows that except for the DS Bowl dataset, our approach overall performs the best in terms of Dice metric. This is the only dataset where our approach achieves lower scores when adding noise, while it outperforms the other block types for the other datasets, we show that it is less sensitive to noise than the other block types in most of the cases.

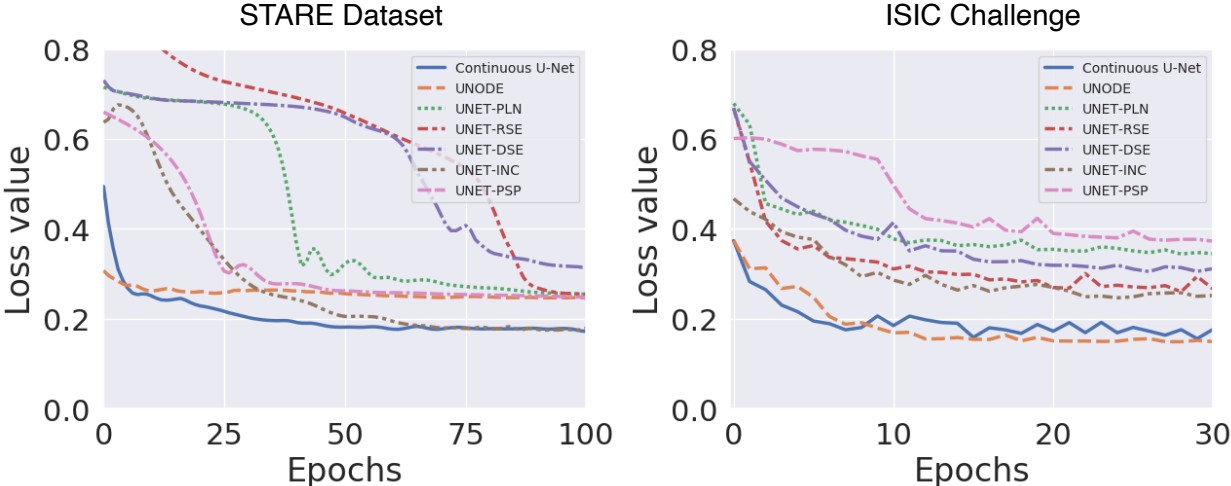

Figure 4: Convergence plots for different types of U-Net blocks on remaining two datasets. The behaviour of our dynamic blocks is denoted in blue colour, where only a few iterations are needed to converge whilst existing blocks need far more iterations.

## A.2   Supplementary Solver Results

In addition to the experimental results displayed in the main paper, we presented some further experiments for the different type of solvers and its effect in our model, displayed in Table 7. The results show that the

fourth order Runge-Kutta solver performs the best for the BUCSI dataset in terms of accuracy and Average Hausdorff distance. For the particular case of the GlaS dataset, however, the implicit Adam-Bashforth-Moulton solver outperforms the RK4 one. Our theroretical insights reported in the main paper follows the empirical findings showing that RK4 is theoretically and empirically is the best choice.

Table 7: Ablation study on different types of ODE solvers on the two remaining medical image segmentation datasets. The best results are highligthed in green colour.

| Solver Type | GlaS Challenge | | | BUSI Dataset | | |
|---|---|---|---|---|---|---|
| | Dice↑ | Acc↑ | AHD↓ | Dice ↑ | Acc↑ | AHD↓ |
| Euler | 0.7141 | 0.7374 | 14.01 | 0.6612 | 0.8957 | 3.36 |
| AB | 0.8455 | 0.8668 | 9.32 | 0.8127 | 0.9419 | 2.28 |
| ABM | 0.8521 | 0.8696 | 9.27 | 0.8203 | 0.9409 | 2.34 |
| RK4 | 0.8469 | 0.8675 | 9.31 | 0.8090 | 0.9447 | 2.15 |

## A.3 Supplementary Performance Comparison & Training Scheme

We provide further numerical comparison of our model against other existing U-type Networks using additional mechanisms. The results are displayed in Table 9 using the DS Bowl dataset, BUSI dataset and ISIC challenge. As for the other datasets, our apporach is able to perform as good as most of the state of the art approaches while even outperforming some of them, even though despite we do not use any additional mechanisms. Finally and for sake of clarity, we provide the training scheme we for our experiments, which is summarised in Table 10. We remark that we use the same setting for all experiments for a fair comparison.

Table 8: Performance comparison, in terms of Dice score, on a range of noise levels. We denote our `continuous U-Net` with ★. The best results are highlighted in green colour.

| | | NOISE LEVEL EXPERIMENTS | | | | | | | | | | | |
|---|---|---|---|---|---|---|---|---|---|---|---|---|---|
| | Block | GlaS Challenge | | | | STARE Dataset | | | | DS Bowl Dataset | | | |
| | Type | 0.0 | 0.2 | 0.4 | 0.5 | 0.0 | 0.2 | 0.4 | 0.5 | 0.0 | 0.2 | 0.4 | 0.5 |
| U-Net | PLN | 0.7616 | 0.5898 | 0.4209 | 0.3977 | 0.7828 | 0.5554 | 0.4929 | 0.4840 | 0.9156 | 0.7635 | 0.6563 | 0.6253 |
| | RSE | 0.7893 | 0.4813 | 0.5133 | 0.5117 | 0.7731 | 0.4805 | 0.4805 | 0.4805 | 0.9131 | 0.5015 | 0.4892 | 0.4863 |
| | DSE | 0.7264 | 0.4564 | 0.3713 | 0.3613 | 0.7358 | 0.4805 | 0.4805 | 0.4805 | 0.9181 | 0.5307 | 0.4842 | 0.4698 |
| | INC | 0.7962 | 0.4517 | 0.4485 | 0.4697 | 0.7750 | 0.5290 | 0.4895 | 0.4829 | 0.9210 | 0.5172 | 0.4849 | 0.4751 |
| | PSP | 0.5523 | 0.5523 | 0.5515 | 0.5469 | 0.4805 | 0.4805 | 0.4806 | 0.4805 | 0.9060 | 0.4895 | 0.4690 | 0.4669 |
| | FO | 0.8045 | 0.5345 | 0.4845 | 0.4801 | 0.5112 | 0.4854 | 0.4788 | 0.4761 | 0.8855 | 0.4844 | 0.4745 | 0.4700 |
| ★ | DB | 0.8469 | 0.8063 | 0.5882 | 0.4922 | 0.8378 | 0.5145 | 0.5034 | 0.5017 | 0.9335 | 0.4939 | 0.4678 | 0.4644 |

Table 9: Performance results for state of the art U-Net architectures with additional mechanisms. We mark our `continuous U-Net` with ★.

| TECHNIQUE | MECHANISM | DS Bowl Dataset | | | BUSI Dataset | | | ISIC Challenge | | |
|---|---|---|---|---|---|---|---|---|---|---|
| | | Dice↑ | Acc↑ | AHD↓ | Dice↑ | Acc↑ | AHD↓ | Dice↑ | Acc↑ | AHD↓ |
| Attn. U-Net | Attn. Gates | 0.9388 | 0.9760 | 3.16 | 0.7484 | 0.9208 | 3.01 | 0.8987 | 0.9331 | 4.98 |
| DynU-Net | Heuristic-R + Opt | 0.9446 | 0.9771 | 3.07 | 0.7716 | 0.9270 | 2.83 | 0.9105 | 0.9408 | 4.48 |
| U2Net | Nested U-Nets | 0.4776 | 0.8723 | 5.97 | 0.5096 | 0.8696 | 4.12 | 0.8310 | 0.8863 | 7.83 |
| UNeXt | Tokenised MLP | 0.6671 | 0.8943 | 5.54 | 0.8064 | 0.9398 | 2.26 | 0.9202 | 0.9482 | 4.01 |
| TransUnet | Transformers | 0.9351 | 0.9760 | 3.17 | 0.8507 | 0.9497 | 2.00 | 0.9251 | 0.9529 | 3.74 |
| ★ | ✗ | 0.9335 | 0.9745 | 3.29 | 0.8090 | 0.9447 | 2.15 | 0.9094 | 0.9433 | 4.46 |

Table 10: Overview of training settings for all experiments.

| Parameter | Value |
|---|---|
| Loss function | Binary Cross Entropy Loss |
| Optimiser | Adam |
| Learning rate | $10^{-4}$ |
| Learning rate schedule | Multiplication of learning rate with 0.999 every epoch |
| Epochs | 500 |
| Batch size | 16 |
| Levels of U-Net architecture | 4 |
| Number of filters per block | 3, 6, 12, 24 |
| Tolerance (for contin. blocks only) | $10^{-3}$ |

