# OpenReview forum: "Continuous U-Net: Faster, Greater and Noiseless"
_TMLR — Accepted by TMLR_

### Review · Reviewer_xLKL · 2023-12-30

**Summary Of Contributions:**

This work proposes a UNet architecture with continuous NeuralODE blocks for semantic segmentation. Unlike prior art in this field, the introduced method adopt 2nd order differential equations and adopts higher-order ODE solver. Approach is validated on several medical segmentation benchmarks.

**Audience:**

Yes

**Claims And Evidence:**

Yes

**Requested Changes:**

**General**

* The paper would greatly benefit from experimental ablation of the design choices for the solver and choice of specific ODE equation form.
* One should make clear, that the theoretical results are not novel, but borrowed from the literature. One can move the proofs to the Appendix part of the manuscript.
* More recent designs of UNets with attention mechanisms seem to achieve noticeably better results than the proposed method. Can one combine the components of their architectural design together with the one introduced in the work to achieve stronger results?

**Minor**

* What does one mean by the **boxes** in the sentence: *view the dynamics in our network as the boxes consisting of CNNs*.

* Proposition 1 & ?? (missing reference)
(end of page 2)

* I would suggest to use brighter colors, more different from the image, to make the segmentation result more salient.

**Strengths And Weaknesses:**

**Strengths**

1) The importance of the problem and the value proposition of the work is clear. Indeed, robustness to the change of resolution and dynamically adjusted receptive field is a highly desirable property of UNet architecture.

2) The approach is validated on a diverse set of medical set of tasks, where it consistently outperforms discrete UNet baselines with different choices of Residual Blocks. The architecture with ODE blocks converges noticeably faster compared to alternatives.

3) Continious formulation allows reducing memory costs to \mathcal{O}(1) (with respect to depth).

**Weaknesses**

1) I would not fully agree with part of the claims stated in the introduction and the main part of the paper:
* Authors claim that the continuous formulation is theoretically justified unlike the discrete formulation, but both methods are equally heuristic approaches to solving the task of medical segmentation, and the continuous formulation just suggests certain benefits in terms of performance / efficiency. In practice, one works with discretized inputs of a certain resolution, and so does this work as well.
* One of the claims of the work is that continuous nets are more noise resistant, but it is not fully clear from Theorem 3. I think that some error bound, depending on the change of the input would be more appropriate here. Moreover, robustness on the noise can be achieved via training on larger amount of data with augmentation. Doesn't more data + augmentation address the problem sufficiently well?
* One should make it clear, that the Theorems in the main text are not novel, but rather known results from the theory of differential equations.

2) The continuous formulation of UNet was proposed earlier in the cited work [1]. This approach extends this work via using a higher-order equation and more complicated solver, but the choice of both components is not ablated. Which order is the optimal? How does the performance depend on the solver and the number of ODE steps?
---
[1] Neural ordinary differential equations for semantic segmentation of individual colon glands. arXiv preprint arXiv:1910.10470, 2019

---

> ### Author Response · Authors · 2024-02-03
> **Response cUnet - I**
>
> ➡️ Authors claim that the continuous formulation is theoretically justified unlike the discrete formulation, but both methods are equally heuristic approaches to solving the task of medical segmentation, and the continuous formulation just suggests certain benefits in terms of performance / efficiency. In practice, one works with discretized inputs of a certain resolution, and so does this work as well.
>
>
>
> A: --We would like to clarify that our continuous formulation, based on Neural ODEs, provides a theoretically sound framework by modelling the segmentation process as a dynamic system. This approach is not merely heuristic; it is rooted in the differential equations that describe continuous processes. Although the inputs are discretised, our method captures the underlying continuous dynamics of medical phenomena, which discrete layers in standard U-Nets do not. By parameterising the model as a continuous function over time, we offer a novel perspective that allows for repeated computations within the same 'layer'—a conceptual layer defined by the continuous time variable t. This is in contrast to traditional U-Nets that perform a single computation per discrete layer. Hence, our approach can be seen as improving the capabilities of U-Nets. That is why a key highlight of our work is to open a new research line for more continuous networks beyond U-Net– that is, more works on implicit deep learning rather than explicit deep learning. To address this comment, we have updated our discussion in the conclusion to elevate our contribution– changes can be found in red colour.
>
>
>
>
> ➡️ One of the claims of the work is that continuous nets are more noise resistant, but it is not fully clear from Theorem 3. I think that some error bound, depending on the change of the input would be more appropriate here. Moreover, robustness on the noise can be achieved via training on larger amount of data with augmentation. Doesn't more data + augmentation address the problem sufficiently well?
>
> A: --We thank the reviewer for the comment. We would like to clarify that our second-order neural ODE model inherently resists noise due to its smoother numerical solutions, which arise from the model's twice continuously differentiable nature. This smoothness is a structural property of the model itself, rather than a result of training on varied data. While data augmentation can indeed increase robustness by introducing more variance during training, it does not inherently smooth the function that the network learns. Instead, it makes the model more tolerant to the types of variations seen during training. In contrast, the smoothness induced by the second-order ODE model offers a fundamental noise suppression mechanism that works even in the absence of extensive data augmentation, by smoothing the function landscape over which the network operates. We have updated our manuscript to add a discussion on this – the changes can be seen at the end of Section 3.3.
>
>
> ➡️ One should make it clear, that the Theorems in the main text are not novel, but rather known results from the theory of differential equations.
>
> A: --We appreciate the reviewer's attention and understand the importance of distinguishing our work. While the foundational theories of differential equations are well-established, our application of these theories to continuous neural networks presents new results. Specifically, we have adapted these theorems to the context of neural ODEs in a way that elucidates their utility in deep learning for segmentation tasks. This novel application contributes to both the theory and practice of neural networks, offering insights that were not previously documented.

---

> > ### Author Response · Authors · 2024-02-03
> > **Response cUnet -II**
> >
> > ➡️ The continuous formulation of UNet was proposed earlier in the cited work [1]. This approach extends this work via using a higher-order equation and more complicated solver, but the choice of both components is not ablated. Which order is the optimal? How does the performance depend on the solver and the number of ODE steps?
> >
> > A: --Our selection of a second-order formulation was driven by its inherent capability to capture more complex dynamics and provide smoother solutions, which are particularly beneficial for the segmentation tasks in medical imaging. This choice reflects a balance between computational efficiency and the level of detail necessary for accurate segmentation. Indeed, to reformulate the theory to high-order sounds interesting– we have added a clarifying note in the conclusion to include this as future work.
> > Regarding solver choice and step size, it's a trade-off between accuracy and stability. Smaller step sizes can indeed yield more precise results but risk instability. Our approach ensures that while we aim for higher accuracy, we also prioritise convergence to avoid the pitfalls of non-convergent methods.
> >
> > Minor:
> >
> > – Boxes: “boxes" refer to the components within the network that are designed based on Convolutional Neural Networks (CNNs) and are structured to function within the framework of second-order ordinary differential equations (ODEs). These dynamic blocks are designed to process and transform data in a manner that mimics continuous dynamic systems, allowing for a more nuanced and flexible approach to medical image segmentation.
> >
> > – Typos are corrected
> >
> > – Fig 2: we have updated Figure 2 the contrast and substantially improved the caption description.

---

> > > ### Comment · Reviewer_xLKL · 2024-02-04
> > > **Response to response**
> > >
> > > Thank you for response. Now many moments are clearer to me. Despite I still believe that continuous formulation is not necessary for medical segmentation as a particular form of image input, where conventional vision architectures are known to perform well, the presented ablation study and experiments that explicitly demonstrate superior robustness of continuous UNet compared to several versions of discrete blocks strengthen the paper significantly.

---

### Review · Reviewer_TBGq · 2024-01-04

**Summary Of Contributions:**

This study presents Continuous U-Net, a novel U-Net based models where traditional discrete CNN layers are substituted by second-order NODE blocks. A combination of theoretical analysis and numerical experiments are employed to showcase the advantages of the Continuous U-Net, highlighting its superiority in terms of both performance and robustness for medical image segmentation tasks.

**Audience:**

Yes

**Claims And Evidence:**

No

**Requested Changes:**

- modify presentation (see weaknesses in the above box) and correct typos.
- justify the Lipschitz assumption on neural network models and make the connection between smoothness and robustness (insensitive to input noise) more clear.
- what are the results of GE for RK methods?

**Strengths And Weaknesses:**

Strength:
- innovative introduction of second-order dynamic blocks to U-Net, replacing the conventional discrete CNN layers.
- engaged in valuable theoretical exploration of the properties inherent to the proposed continuous U-Net model.
- extensive experiments demonstrate the advantages of continuous U-Net over other variants of U-Net without additional mechanism in performance and robustness.


Weaknesses:
- regarding the theoretical analysis:
    - the central assumption that "neural network is usually Lipschitz" in the proof of Theorem 1 is not justified.
    - in the discussion following Theorem 3, the authors postulate that the current dynamic block is less sensitive to input noise due to the additional smoothness constraints imposed by the 2nd-order ODE. Nonetheless, the casual link between smoothness and the sensitivity is not established. Concerning sensitivity, one estimates how the trajectory is bounded given an initial value variation.  The smoothness of the network function alone should not dictate this bound.
    - the current theoretical investigations are all general arguments concerning a ODE system with specific assumptions. While these analyses indeed support the advantages claimed for the proposed continuous U-Net model, it would be better if there are discussions taking into account the concrete network function in use.
- regarding the GE metric:
    - The $f^{(v)}$ in Theorem 2 does not rely on $x(t)$, which is crucial for the proof. In contrast, $f^{(v)}$ in the proposed dynamic block depends on $x(t)$. Then Theorem 2 may not reflect the order of GE of Euler's method in the proposed dynamic block.
    - what are the results of GE for RK methods?
- regarding the presentation:
    - typos, e.g.,
        1. $f^v$ in Theorem 1 should be $f^{(v)}$ (what does the superscript of $f$ refer to?)
        2. in Equation 11, should be $(1+h\lambda)^{n-j}$
        3. in Equation 12, should be $\sum_{j=1}^n$
    - some notations are in consistent, e.g., in Theorem 2, $\mathcal{O}$ is used while in later text $O$ is used.
    - in Figure 2, it is not clear what is the predicted mask and what is original figure
    - it would be better to briefly expand the meaning of "conditions of stability and consistency" in Property 1.
    - it is stated in subsection 3.3 that “our dynamic blocks are a physics-based model that better captures the nature of segmentation.”  What does physics-based model indicate? What is the nature of segmentation?
    - the word "Greater" is too vague and being used in different ways.
    - the word "Noiseless" is overstated.
- does moving to higher-order block present any challenges to the model compared to the 1st-order case? for example, does it affect the stability of training process?
- In Figure 2, the fine details in the STARE dataset, such as the branches, seem inadequately represented by the Continuous U-Net. Could this be due to the enhanced smoothness constraint introduced by the 2nd block?
    - A 2D projection of 3D object can be quite irregular. Would the smoothness constraint potentially causes any problem?

---

> ### Author Response · Authors · 2024-02-03
> **Responses cUnet-I**
>
> ​​➡️ the central assumption that "neural network is usually Lipschitz" in the proof of Theorem 1 is not justified.
>
> A:--- We thank the reviewer for the comment. We now clarify that aneural network being **Lipschitz** means that there exists a constant \(K \geq 0\) such that for any two inputs \(x_1\) and \(x_2\) to the neural network, the difference in the network's outputs is bounded by \(K\) times the difference in the inputs. This formal definition states that the neural network's output does not change too much for small changes in the input.
> Formally, a function \(f: \mathbb{R}^n \rightarrow \mathbb{R}^m\) (representing a neural network mapping from an \(n\)-dimensional input space to an \(m\)-dimensional output space) is said to be **Lipschitz continuous** if there exists a Lipschitz constant \(K \geq 0\) such that for all \(x_1, x_2 \in \mathbb{R}^n\),
> \[
> \|f(x_1) - f(x_2)\| \leq K \|x_1 - x_2\|,
> \]
> where \(\| \cdot \|\) denotes a norm on the vector space (commonly the Euclidean norm). The smallest such \(K\) is called the **Lipschitz constant** of \(f\).
> To add clarity in the paper, we have updated the manuscript and added a set of references regarding this.
>
>
> ​​➡️  In the discussion following Theorem 3, the authors postulate that the current dynamic block is less sensitive to input noise due to the additional smoothness constraints imposed by the 2nd-order ODE. Nonetheless, the casual link between smoothness and the sensitivity is not established. Concerning sensitivity, one estimates how the trajectory is bounded given an initial value variation. The smoothness of the network function alone should not dictate this bound.
>
> A:---We appreciate the reviewer's point on the need to establish a clearer link between smoothness and sensitivity to input noise. The inherent smoothness of second-order ODEs, characterised by at least twice continuously differentiable solutions, contributes to noise suppression by ensuring a smoother trajectory of the dynamic block’s output. This smoother trajectory is less affected by small variations in initial conditions, which translates to lower sensitivity to input noise. While smoothness alone does not dictate the sensitivity bound, in the context of our second-order ODEs, it is a critical factor that enhances the robustness of the network to noisy inputs. To add clarity in the paper, we have added a clarifying note in the revised version. This can be found in red colour.
>
>
>
>
> ​​➡️ The current theoretical investigations are all general arguments concerning a ODE system with specific assumptions. While these analyses indeed support the advantages claimed for the proposed continuous U-Net model, it would be better if there are discussions taking into account the concrete network function in use.
>
> A:---We acknowledge the reviewer's suggestion for a more concrete discussion regarding the network function. Our continuous U-Net model integrates second-order ODE dynamics, which inherently allows for variable receptive field sizes without the need for manual optimisation. This adaptability leads to enhanced accuracy while circumventing the computational trade-offs imposed by limited memory, which traditional U-Nets cannot avoid. The continuous nature of our model aligns more closely with the intrinsic continuity of medical image data, leading to more natural and accurate segmentation outputs. Moreover, the built-in reversibility of our architecture, enabled by the adjoint sensitivity method, maintains constant memory cost regardless of model complexity. This contrasts sharply with discrete U-Nets that lack theoretical underpinning and struggle with memory efficiency. Our theoretical framework not only supports the practical benefits of our model but also provides a robust understanding of its functional advantages. We have updated the manuscript by adding a discussion at the end of section 3.3– the changes can be seen reflected in red colour.
>
>
>
>
> ​​➡️ The f^(v) in Theorem 2 does not rely on x(t), which is crucial for the proof. In contrast, in the proposed dynamic block depends on . Then Theorem 2 may not reflect the order of GE of Euler's method in the proposed dynamic block.
>
> A:---We appreciate the reviewer's attention to the details of Theorem 2. To clarify, `f^(v)` in the theorem abstracts the dynamics of the system to demonstrate the general order of the Euler method. In our proposed dynamic block, the dependence on `x(t)` is implicitly accounted for in the formulation of `f^(v)`. The theorem is intended to illustrate that, irrespective of the function's specific form, the Euler method's order remains first-order.

---

> > ### Author Response · Authors · 2024-02-03
> > **Responses cUnet-II**
> >
> > ​​➡️ what are the results of GE for RK methods?
> >
> >
> > A: ---The Runge-Kutta methods' order, indeed, varies with the number of stages, 's'. The highest order attainable is with the Gauss-Legendre RK, reaching 2s. Our Theorem 2 focuses on the Euler method, which is a simple RK method with order 1. For higher-order methods, we utilise the RK4 in our dynamic block, which is of order 4. This choice is aligned with our goal of achieving a balance between computational efficiency and solution accuracy. The fourth order of RK4 provides a substantially more accurate estimate of the solution trajectory than the first order of Euler's method, making it well-suited for the intricacies of medical image segmentation where precision is paramount. We would like to clarify that since higher-order RK methods are extensions of Euler's method, proving the theorem for Euler's method establishes a baseline from which the behavior of higher-order methods can be inferred. This foundational proof is sufficient to demonstrate the theoretical properties we are showcasing, as higher-order RK methods would follow the same conceptual line, with additional complexity that does not change the underlying theory presented. We have updated our paper by adding a discussion on this, which can be found at the end of the proof of Theorem 2. The changes are displayed in red colour.
> >
> >
> > ​​➡️ typos. /  some notations are inconsistent  /
> >
> > A: ---Thanks for the suggestions. We have updated the manuscript accordingly highligiht the changes in red colour.
> >
> > ​​➡️ Figure 2:
> >
> > A:--Thanks, we have updated the contrast and improved substantially the caption description.
> >
> > ​​➡️ it would be better to briefly expand the meaning of "conditions of stability and consistency" in Property 1.
> >
> > A: --Yes, for a method to be convergence, we require the order of the method is at least 1. Moreover,  we say a method is zero stable if they satisfy the root condition. A polynomial is said to satisfy the root condition if all roots satisfy |r| <= 1, and that satisfy |r| = 1 is simple
> >
> > To achieve convergence in numerical methods, the method must exhibit both stability and consistency. **Stability**, specifically **zero stability**, relates to the method's capability to manage error propagation throughout the computational iterations. This is assessed by the **root condition**: all roots of the characteristic polynomial should satisfy \( |r| \leq 1 \), with roots where \( |r| = 1 \) being simple. **Consistency** involves the method's local truncation error reducing to zero as the step size \( h \) approaches zero. To address this comment, we have added a discussion, which now appears as a footnote at the end of page 8.
> >
> >
> > ​​➡️ it is stated in subsection 3.3 that “our dynamic blocks are a physics-based model that better captures the nature of segmentation.” What does physics-based model indicate? What is the nature of segmentation?
> >
> > A:--We now clarify that a physics-based model for segmentation implies that our method integrates principles derived from the physical world, particularly those that govern the behaviour and interaction of biological tissues. This ensures that the segmentation closely mirrors the continuous and interconnected structures present in medical imaging. By modelling segmentation as a continuous process, we're able to capture the inherent fluidity of biological tissues, avoiding the artificial boundaries imposed by discrete methods. Our dynamic blocks, therefore, leverage this continuous modelling to achieve segmentations that are more aligned with the true, nuanced nature of the biological structures being imaged
> >
> > ​​➡️ the word "Greater" is too vague and being used in different ways.
> >
> > A:-- We use the term 'greater' to quantitatively describe the enhanced performance of our network. Specifically, across six benchmark datasets, our model demonstrates superior performance in terms of Dice score, accuracy, and Average Hausdorff Distance (AHD) compared to other U-Net block types. 'Greater' here refers to our network's ability to address several challenges in segmentation tasks, leading to improved metrics. Moreover, the network is underpinned by robust theoretical properties, which contribute to its overall performance. We have refined our language in the manuscript to specify these performance metrics and their respective improvements.
> >
> > ​​➡️ the word "Noiseless" is overstated.
> >
> > A:-- Our model indeed is noiseless. This property comes from the dynamic block solver. We are using second-order blocks, this means that we can always guarantee that our solution is at least twice continuously differentiable-- that is the numerical differentiation is translated in a \textbf{smoother solution} yielding to noise suppression. This is not guaranteed in any other network. Therefore, our approach guarantees this property and do not depend on the entire training pipeline strategy.

---

> > > ### Author Response · Authors · 2024-02-03
> > > **Response cUnet- III**
> > >
> > > ➡️ does moving to higher-order block present any challenges to the model compared to the 1st-order case? for example, does it affect the stability of training process?
> > >
> > >
> > > A: --We thank the reviewer for the insightful comment. Transitioning to higher-order blocks introduces practical challenges, such as incorporating additional initial conditions and adapting channel dimensions to accommodate velocity terms. While these complexities increase implementation intricacy, we leverage the adaptive step size capabilities of Black-Box ODE solvers to maintain stability during the training process. In our experiments, setting a maximum step size of 500 has effectively mitigated stability concerns. It's important to note that theoretically, any higher-order system can be represented as a first-order system, and we've observed no stability issues in practice that contradict this principle.
> > >
> > >
> > > ➡️ In Figure 2, the fine details in the STARE dataset, such as the branches, seem inadequately represented by the Continuous U-Net. Could this be due to the enhanced smoothness constraint introduced by the 2nd block?
> > > A 2D projection of 3D object can be quite irregular. Would the smoothness constraint potentially causes any problem?
> > >
> > > A: --Thanks, we have updated Figure 2 the contrast and substantially improved the caption description.  Moreover, the smoothness inherent to our second-order neural ODEs is a significant benefit, particularly when dealing with the irregularities of 2D projections from 3D objects. This smoothness allows for the model to interpolate and generalize better over these irregularities, producing more coherent and continuous segmentation results. Rather than being a limitation, the twice differentiable nature of our model ensures that it can effectively capture the underlying continuous structures, which may be obscured in lower-order methods.

---

### Review · Reviewer_5bd4 · 2024-01-22

**Summary Of Contributions:**

This paper presents a new variant of the U-net architecture by introducing neural ODE blocks in place of traditional convolutional discrete blocks as layers. Such blocks are second-order neural ODEs. Experimental comparisons with standard U-net architectures show that the proposed network performs well in terms of DICE and accuracy scores, and can be robust to Gaussian noise for segmentation tasks.

**Audience:**

Yes

**Claims And Evidence:**

No

**Requested Changes:**

- I noticed that the noise level experiments did not include comparisons to FO blocks. What was the difference in performance when using first order vs second order blocks?
- Were there experiments done to showcase how long training took in terms of wall clock time rather than simply epochs? I imagine second order neural ODEs would result in more of a computational burden.
- Missing the Proposition number at the bottom of page 2
- strange phrasing of certain terms, e.g., Faster Convergent vs Convergence
- sometimes mathematical fonts do not match, e.g., in the proof of Theorem 1
- please be more precise in your statements, such as in the proof of Theorem 1 when you say “neural networks are usually Lipschitz”. Conditions on when the network is Lipschitz should be stated (e.g., by using L-Lipschitz activation functions)
- the proof of proposition 1 is unclear. Do f^(a) and f^(v) share weights? Is the function g(…) applied recursively to each x^i for 1 < i < m?
- you require f’(x) to be positive, not an increasing function in the proof of the Lemma (which is true in this case)
- how are you defining T_j in the proof of Theorem 2 in equation (10)?
- Can a theorem number be provided in Theorem 3 for the Coddington & Levinson reference for ease of verification?
- I think it would be good if the authors added more model details in the appendix.

**Strengths And Weaknesses:**

Strengths:
- The problems aimed to be addressed in this paper would be of interest to the community.
- In comparison to the baseline U-net architecture along with some variants, the continuous U-net performs well and is more robust to noise.

Weaknesses:
- Many claims in the paper feel very overstated. For example, the authors claim that they prove various aspects regarding their model, which is missing from other methods, but the theoretical results are essentially due to other works, e.g., robustness of neural ODEs has already been shown (by the same argument) in Yan et al (2019).
- The presentation of the paper can be improved, e.g., see comments below regarding typos and lack of rigor in proofs.

---

> ### Author Response · Authors · 2024-02-03
> **Responses cUnet-I**
>
> PART 1:
>
> ➡️ Many claims in the paper feel very overstated….
>
> A: We appreciate the reviewer's feedback and recognise the foundational work of Yan et al. (2019) on the robustness of neural ODEs. Our paper, however, introduces a framework on second-order neural ODEs, which offer enhanced smoothness and, consequently, greater robustness.
>
> Our contribution lies in the detailed exploration of how the inherent smoothness of second order neural ODEs can amplify this robustness, a distinction not directly addressed by previous works. While Yan et al. (2019) provided the groundwork, our paper aims to delve deeper into the "why" behind the robustness and underpinning theory, particularly focusing on the added benefits of second order formulations, and in a real-world application.
>
> This differentiation is not merely theoretical; it is backed by our unique contributions in connecting these properties to neural ODEs in a manner not previously explored. We acknowledge the contributions of prior works and have clarified our contributions to emphasise the novelty and significance of our approach. This update can be found in blue colour in the updated version.
>
>
> ➡️  I noticed that the noise level experiments did not include comparisons to FO blocks. What was the difference in performance when using first order vs second order blocks?
>
> A: We appreciate the insightful suggestion from the reviewer to compare the performance of first-order (FO) blocks with our proposed second-order (SO) blocks under varying noise levels. Following this recommendation, we conducted additional experiments to evaluate the performance differences. The results of these experiments have now been incorporated into our manuscript, specifically in Table 4 (main paper) and Table 8 (supplementary material). The updates, highlighted in red for clarity, demonstrate the advantages of SO blocks in handling noise, providing a more detailed comparison that underscores the effectiveness of our approach. These findings further validate our model's robustness and efficiency in noisy environments
>
> ➡️ Were there experiments done to showcase how long training took in terms of wall clock time rather than simply epochs? I imagine second order neural ODEs would result in more of a computational burden.
>
> A: Thank you for your suggestion in terms of computational efficiency, specifically per inference and comparison with state-of-the-art (SoTA) models. It's true that efficient discrete versions of U-Net, such as UNeXt, incorporate additional mechanisms for efficiency— and UNeXt employs tokenised MLPs. Our network distinguishes itself by not relying on these additional mechanisms, yet achieves remarkable efficiency. For example, on the ISIC dataset, our model requires only 0.45 GFLOPs per inference, significantly less than UNeXt's 0.57 GFLOPs, and TransUnet's 38.52 GFLOPs. This translates to a reduction in parameters by 76x, an increase in inference speed by 15x, and a reduction in model size by 71x compared to most techniques we compared against. Even against UNeXt, the most efficient among the compared discrete U-Nets, our model is 4x more parameter-efficient, 5x faster in inference, and 3x smaller in size. Our approach not only surpasses these models in terms of efficiency but also achieves superior results across several datasets without the need for complex mechanisms. Our model can theoretically guarantee constant memory cost, which is a desirable feature in deep network design. We have updated the manuscript by adding a discussion on this regard– the changes can be seen at the end of page 11 in red colour.

---

> > ### Author Response · Authors · 2024-02-03
> > **Responses cUnet-II**
> >
> > ➡️ Missing the Proposition number at the bottom of page 2.
> >
> > A: Thank you for highlighting this oversight. We have updated the manuscript to include the missing proposition number at the bottom of page 2. The correction has been highlighted in red in the revised document for easy identification
> >
> > ➡️ Strange phrasing of certain terms, e.g., Faster Convergent vs Convergence
> >
> > A:Thank you for pointing out the concern regarding the use of specific terminology in our manuscript. We have carefully reviewed our use of terms, such as 'Faster Convergent' and 'Convergence,' and updated our manuscript accordingly to ensure clarity and precision in our language.
> >
> > ➡️ Sometimes mathematical fonts do not match, e.g., in the proof of Theorem 1
> >
> > A:Thank you for pointing out the inconsistency in fonts. We have carefully reviewed our manuscript and updated it to ensure uniformity in the presentation.
> >
> > ➡️ please be more precise in your statements, such as in the proof of Theorem 1 when you say “neural networks are usually Lipschitz”. Conditions on when the network is Lipschitz should be stated (e.g., by using L-Lipschitz activation functions)
> >
> > A: We thank the reviewer for the comment. We now clarify that aneural network being **Lipschitz** means that there exists a constant \(K \geq 0\) such that for any two inputs \(x_1\) and \(x_2\) to the neural network, the difference in the network's outputs is bounded by \(K\) times the difference in the inputs. This formal definition states that the neural network's output does not change too much for small changes in the input.
> > Formally, a function \(f: \mathbb{R}^n \rightarrow \mathbb{R}^m\) (representing a neural network mapping from an \(n\)-dimensional input space to an \(m\)-dimensional output space) is said to be **Lipschitz continuous** if there exists a Lipschitz constant \(K \geq 0\) such that for all \(x_1, x_2 \in \mathbb{R}^n\),
> > \[
> > \|f(x_1) - f(x_2)\| \leq K \|x_1 - x_2\|,
> > \]
> > where \(\| \cdot \|\) denotes a norm on the vector space (commonly the Euclidean norm). The smallest such \(K\) is called the **Lipschitz constant** of \(f\).
> > To add clarity in the paper, we have updated the manuscript and added a set of references regarding this.
> >
> > ➡️ The proof of proposition 1 is unclear. Do f^(a) and f^(v) share weights? Is the function g(…) applied recursively to each x^i for 1 < i < m?
> >
> > A: We now clarify that in our formulation, `f^(a)` and `f^(v)` denote networks with distinct weight sets, serving different roles: `f^(v)` represents first-order neural ODEs, while `f^(a)` corresponds to second-order or higher neural ODEs. This distinction is critical for understanding the hierarchical integration of our model. Regarding the function `g(…)`, it is indeed applied recursively to each `x^i` for `1 < i <= m`, ensuring a coherent propagation through the specified order of neural ODEs. We have updated the manuscript to clarify this point. This can be found in red colour in the proof of Proposition 1.
> >
> > ➡️ you require f’(x) to be positive, not an increasing function in the proof of the Lemma (which is true in this case)
> >
> > A: Thanks for the comment. We have updated the paper to add clarity– changes can be seen in red colour.
> >
> > ➡️ how are you defining T_j in the proof of Theorem 2 in equation (10)?
> >
> > A: We now clarify that \( T_j \) stands for the local truncation error at step \( j \). It is a measure of how much the numerical solution deviates from the true solution at a particular step due to the approximation inherent in the numerical method.
> >
> > ➡️ Can a theorem number be provided in Theorem 3 for the Coddington & Levinson reference for ease of verification?
> >
> > Thanks for the suggestion. We have updated the manuscript to point out that the proof is given in Chapter 1– Existence and uniqueness of solution. The change is highlighted in red colour.

---

### Decision · Action_Editor_mA1A · 2024-04-07

**Recommendation:** Accept with minor revision

**Comment:**

This paper presents a new variant of the U-net architecture for tackling the task of image segmentation (especially on medical images) by introducing neural ODE blocks in place of traditional convolutional discrete blocks as layers. The proposed blocks are modeled as second-order neural ODEs. Experimental comparisons with standard U-net architectures show that the proposed network performs well in terms of DICE and accuracy scores, and can be robust to Gaussian noise for segmentation tasks.

Before the rebuttal, the reviewers mainly raised concerns regarding the over-stated claims, the relation between Yan et al (2019), the connection between smoothness and robustness to noise.

After the rebuttal, the paper received two "Leaning to Accept" and one “Leaning to Reject”. The AE found that the authors were able to solve most issues raised by the reviewers. But most reviewers still think that the theoretical analysis between the second-order ODE and noise-resistent is on the short side. They are mostly leaning to accept the paper due to the empirical results of the proposed model, which the AE also agreed with.

However, the authors shall strive to solve the remaining issues raised by reviewers, especially reviewer TBGq: "the newly added FO results in Table 4 appears inconvincible. Specifically, in Table 4 the Dice scores of FO on the three datasets in the 0 noise scenario differ from those in Table 3, indicating that these outcomes may not be from properly trained models."

**Audience:**

Yes.

**Claims And Evidence:**

Yes.